# We Can Define the Domain of Information Online and Thus Globally Uniformly

## Wolfgang Orthuber

Department of Orthodontics, UKSH University Hospital Schleswig-Holstein, Kiel University, 24105 Kiel, Germany; orthuber@kfo-zmk.uni-kiel.de

**Abstract:** Any information is (transported as) a selection from an ordered set, which is the "domain" of the information. For example, any piece of digital information is a number sequence that represents such a selection. Its senders and receivers (with software) should know the format and domain of the number sequence in a uniform way worldwide. So far, this is not guaranteed. However, it can be guaranteed after the introduction of the new "Domain Vector" (DV) data structure: "UL plus number sequence". Thereby "UL" is a "Uniform Locator", which is an efficient global pointer to the machine-readable online definition of the number sequence. The online definition can be adapted to the application so that the DV represents the application-specific, reproducible features in a precise (one-to-one), comparable, and globally searchable manner. The systematic, nestable online definition of domains of digital information (number sequences) and the globally defined DV data structure have great technical potential and are recommended as a central focus of future computer science.

**Keywords:** big data; efficiency; similarity search; information; selection; online definition; adapted domain; metric space; domain vector; domain space

## 1. Introduction

Global text search engines are indispensable for finding information on the Internet. However, text-based information is only a small part. Quantitative information allows much more precision and objectivity. So far, however, its digital representation is neither globally comparable nor searchable. The digital number sequences that represent quantitative information are defined only locally and by context, i.e., they are not defined in a globally uniform way. Therefore, they are not globally comparable and not searchable.

Digital number sequences are not globally defined, although these are building blocks of all digital information (the digital bits are parts of number sequences). They can represent all kinds of information, including non-trivial, complex digital information. It should therefore be possible to define number sequences (digital information) in a globally uniform way so that they can also be compared and exchanged globally. In the age of the Internet, this would be very efficiently possible. If an appropriate infrastructure is available, it is sufficient to prefix given digital information (number sequence) with the global pointer to its online definition. I have published [1–14] on this and called the resulting data structure "Domain Vector" or "DV" for short. It has the structure:

$$\text{DV: UL plus number sequence} \tag{1}$$

Here, "UL" is a "Uniform Locator" which is an efficient global pointer to the machine-readable, unique online definition of the number sequence. DVs with the same UL are automatically globally defined by the same online definition. Thus, the UL is a global pointer and identifier. It has a similar function as the URL [6] of a conventional link but is optimized for efficiency. Therefore, it is directly represented as a hierarchical number sequence [12]. The binary encoding of DVs can be carried out very efficiently.

But this has not been realized until today! I could hardly believe this at first, after I had started to deal with the topic in detail more than 10 years ago. At first, I wanted to integrate some ideas into existing approaches, e.g., by globalizing the existing database concept [1,2]. There were also interested colleagues from computer science. We wrote a joint publication [3] to find a bridge to large groups focusing on data exchange on the Internet [15–18] to start a discussion about it. I gave talks at conferences and wrote more publications. These come more and more clearly to the point [4–10]. However, there was no relevant response. The problem remains and has existed for decades [11]: We cannot define data (number sequences) globally and then compare, share, and search them globally. So there is a significant gap in the basics, which unfortunately cannot be glossed over.

## 2. Basics and Recommended Method for Global Definition of Digital Information

Although I have written and published much about the new globally defined digital data structure (1) of the DV, this was largely ignored. I am under the impression that there is no motivation to discuss and rethink fundamentals that challenge models of thinking that have been built up over many years. I had superficial contact with some decision-makers in computer science. However, they were not very interested. Apparently, even experts often underestimate the scope of a (language-independent) global definition of digital information (number sequences). Perhaps this has not (yet) been a sufficient focus in computer science. Therefore, a short recapitulation of these basics seems appropriate.

The result of any well-defined physical measurement is "information" [19]. It is transportable and a selection from a well-defined ordered set of all possible measurement results. This set is called "domain" (of information). In short:

$$\text{Information is a selection from a domain} \tag{2}$$

The domain must be clear (known) to the participants of the conversation *before* the information is exchanged. Additionally, any digital information, i.e., every digital bit, number and number sequence, is a selection from a domain (Figure 1).

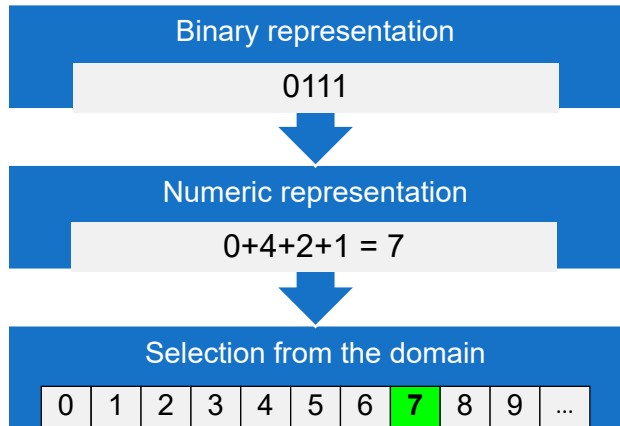

**Figure 1.** Exemplary selection from the domain 0 . . . 15 by 4 bits of digital information. The binary coding and the meaning of each possible number 0 . . . 15 must be clear to the participants of the digital communication.

It is assumed here that the reader knows important basics of set theory and binary representation of information:

- Digital bits first encode numbers (Figure 1). The software must know the binary format of the numbers.
- Each number represents a selection from an ordered set or "domain" (Figure 1).
- A sequence of multiple numbers selects from multiple domains or a "multidimensional" domain (which is ordered along multiple dimensions). It is the domain of the numbers and so the domain of this piece of digital information.

- The elements of the domain can represent very different things, e.g., control codes for software, letters, indices, e.g., on entries of a table, of course also measurable values (Figure 2) and other data.

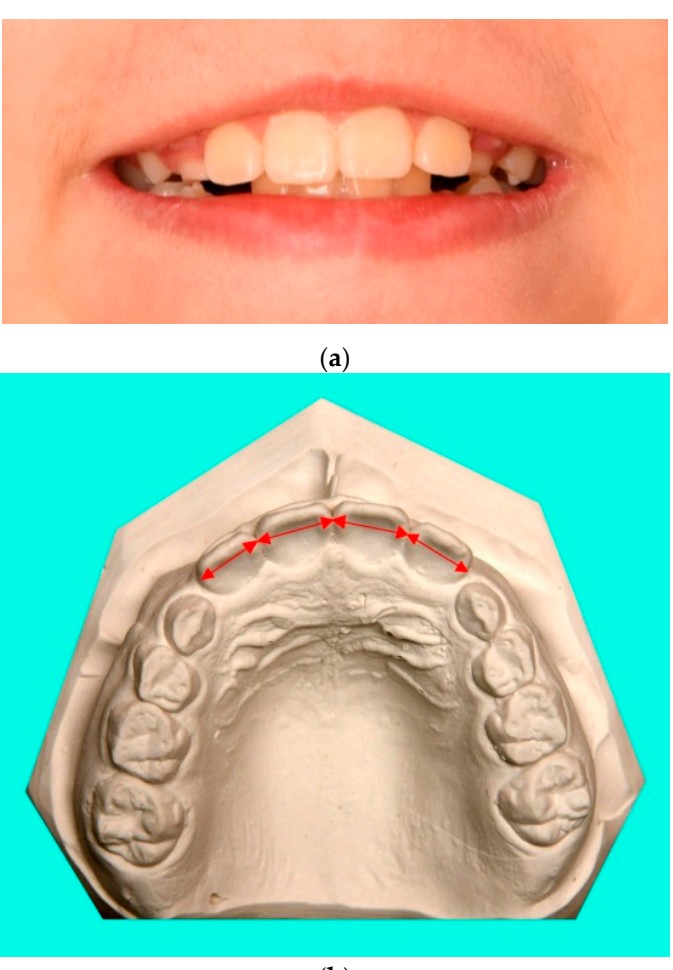

**(a)**

**(b)**

**Figure 2.** (**a**) Correction and preservation of the position of the incisors is part of the orthodontic treatment goals. The dentition model of this patient is shown in (**b**). The red arrows show the sum of the incisor widths. This is used to estimate their space requirements and is an example of an orthodontic measurement. Many such data (number sequences) must be combined for diagnosis and treatment planning. Globally defined data enables direct global comparison and efficient exchange of experience. More details about medical applications are available in former publications (e.g., [5,7,10]).

The global definition of the domain (of the digital numbers and their numerical format) is important to globally obtain the same binary representation of the information and the comparability of ordered (quantitative) data worldwide. The domain is ordered in one or more dimensions so that each element of the domain can be quickly selected by numbers. This ordering also allows the precise definition of a "metric" or "distance function" for reproducible similarity comparison and similarity search [20] of elements (information) in the domain (see Figures 3 and 4). A domain with a metric (a metric space) is called a "domain space" [14]. The count of numbers used to select its elements is the dimensionality of the domain. In the case of more than one number, the domain is "multidimensional" and its size or "cardinality" can be very large. For example, if each number represents a measurement result 0 . . . 99 (only $10^2$ = 100 possibilities per number), then a sequence of six such numbers (e.g., mm coordinates of only 2 points in a 10 cm cube) selects from a domain with $10^{(6 \times 2)}$ or 1000 billion elements. So the size of the domain increases rapidly. It increases exponentially with its dimensionality, i.e., the length of the number sequence. In

the case of multiple measurements, e.g., medical findings, the domain becomes extremely large. If such a domain is unambiguously defined online, it can be a powerful basis for reproducible information transport.

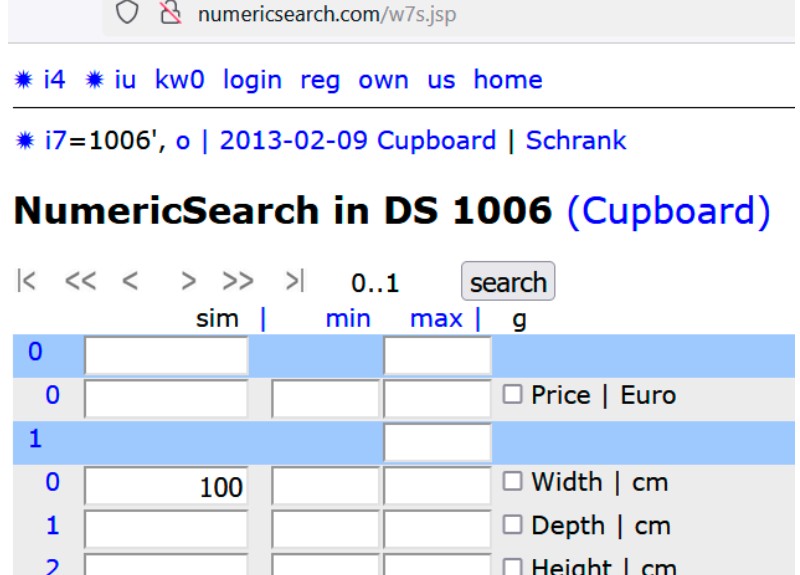

**Figure 3.** Example of a search for cupboards with 100 cm width. The numbers "Price, Width, Depth, Height" were defined as relevant (searchable and comparable) features of a cupboard. These 4 numbers select from a 4-dimensional domain. We can select another domain definition by clicking on "* i7" and view the defined DVs by clicking on "* i4". Open data of the users are viewable after clicking on "* iu".

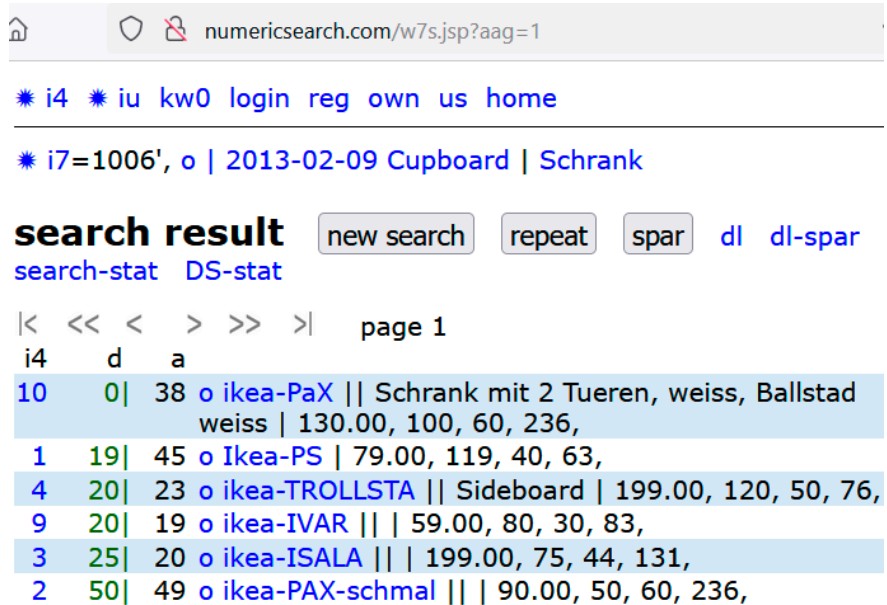

**Figure 4.** Search result after search input of Figure 3. The smaller the absolute difference (d) of the width from 100 cm, the higher the rank in the search result. Therefore, the cupboards whose width (2nd number) is closest to 100 (with the smallest distance (d) from 100) are listed first. Globally (online) defined DVs (1) would allow global searches.

We need to focus our definition on the features that are important for a particular application. These should be reproducible and converted one-to-one (bijectively) into a

digital number sequence. Figure 2 shows an example from medical diagnostics. The sum of the incisor widths shown, together with other measurements, is one of many findings in orthodontics to plan necessary tooth movements and corresponding orthodontic appliances. With a global definition, orthodontists could share objectifiable experiences for similar findings. In doing so, they could directly compare statistics on (measurable) treatment outcomes for different treatment strategies.

The numbers which represent such measured values are selections from domains such as those shown in Figure 1.

Here are some recommendations for the online definition of such number sequences with application-specific adapted domain [9]:

(a)  The original source of information (e.g., observation, measurement) should be reproducible.

(b)  We create a list of all independent, reproducible features (e.g., measurements, properties) that could be of interest to this application.

(c)  We try to search, reference, and reuse existing online definitions for quantifying each listed feature. If no suitable online definition is available, we must create a new online definition of our own. This definition can use and describe the natural order of the feature and select a natural unit, or it can define a new order of feature variants so that they are reproducibly (one-to-one resp. bijectively) mapped to the domain of a number (quantification). Then, they are selectable by the number (which digitally represents the feature). The defined order of application-specific features can later be automatically used for similarity comparison and search by universal numerical search engines [14].

(d)  The digital representations (numbers) of all listed features are uniquely combined and nested to form the number sequence that digitally conveys the information for that application.

(e)  All necessary definitions (of the features, of the representing numbers, of their combination, of the digital format of each number) are put online in standardized machine-readable form. The online definition starts with a version number of the standard and an efficient machine-readable specification of the binary format of the number sequence. This includes short links to the exact definitions of each number. The exact definition may be very detailed and may also include links to multilingual and multimedia explanations and to (the source code of) software for handling the number sequence. Online definitions can be reused and nested in subsequent online definitions. This enables the definition and handling of complex data structures.

(f)  After the online definition, the defined information can be reproducibly digitized and transported as DV in the form (1): "UL plus number sequence", where UL efficiently refers to the online definition.

The first steps (a), (b) and (c) are best solved by a collaboration of trained computer experts with the professionals who will use the online definition later. Certain aspects such as "reproducibility" may be unfamiliar at first, especially in the case of more complex multidimensional information, such as in medicine. It helps to think of the (transportable) components of information and to consider the extent to which the familiar terminology is reproducible and/or how reproducibility and sufficient accuracy can be achieved.

A bijectively (one-to-one) adapted domain of defined numbers to original information is called "*adapted domain*" [9]. It is adapted to the application to allow reproducible application-specific information transport. The online definition of the numbers and their domains is accessible worldwide.

## 3. Example of Results

Section 2 describes the exact and reproducible global digital representation of information. Among other things, this enables the exact and reproducible global search for information.

As mentioned, the definition of number sequences with domains and the associated data search [20] is technically solved. Additionally, for demonstration purposes, I have

programmed a prototype on a local database [14] to motivate the global definition of number sequences.

As an example, Figure 3 shows the definition of four features resp. numbers (price in EUR, Width, Depth and Height in cm) which may be of interest when buying a cupboard. Those DVs (cupboards) with the second number (width in cm) nearest to 100 are searched, and Figure 4 shows the search result. For example, the cupboard listed first has exactly the width of 100cm (second number = 100) and therefore distance d = 0 from the searched value 100. This example is only to illustrate and recall the principles. Users can define the numbers freely, according to their interests. Using the (globally) online defined DV data structure (1), and such definitions could be globally valid.

## 4. Discussion

### 4.1. Short Conclusive Argumentation

Section 3 above, where exemplary results (searchable information) are shown, is for illustrative purposes only and is not necessary for deriving the conclusions. The conclusions of this paper can be directly derived by simple conclusive argumentation, starting from set theory:

- Information (e.g., digital information) is a selection (2) from an ordered set. I have called this set the "domain" of information. It is clear that numbers are selections from ordered sets and that digital bits encode sequences of numbers.
- Therefore, for a direct comparison (e.g., to check similarities in a similarity search), the domain (2) of the compared pieces of digital information (number sequences) should be the same.
- Therefore, it is consequent to consider number sequences as building blocks of digital information and to define the domain of a number sequence and its binary format globally. In the current infrastructure, this can be performed efficiently by a machine-readable online definition of digital information (number sequence).
- The proposed DV data structure (1) simply represents such a number sequence, preceded by the UL, which is required as a global identifier and efficient pointer to the online definition. To save energy and resources, the binary format of the DV (UL and number sequence) is optimized for efficiency.
- Details about a possible format of the UL were published [12]. Additional data are not necessary for the DV and were avoided. This is relevant for both efficiency and global uniqueness. In contrast, the (globally unique) online definition of the DV may contain efficient links to additional detailed descriptions and explanations.

### 4.2. More Detailed Discussion

The online defined DV data structure (1) allows an exact, reproducible search, as shown in the example above (Section 3). With this, the searchable numbers can be freely defined according to the interests of the users. Complex multidimensional global definitions could enable completely new possibilities for global information transport, comparison and searching. These important technical facts are already clear. We will therefore include further aspects in this discussion.

After all the successes of computer science over many years, it may come as a surprise that there is still much room for improvement in the digitization of information. To realize this, it first helps to remember that any information is (transported as) a selection from a set: The domain (2) of information. It is the "set of possibilities" which must be clear *before* information exchange. However, apart from special cases (which are already important, such as Unicode [21]), the globally uniform definition of domains of digital information (sequences of numbers) has not yet been systematically focused on in information science. The far-reaching consequences became apparent to me when I realized what would actually be possible with the (nestable, global) online definition of such domains. Therefore, I searched the Internet for the term "information" and I found an extensive discussion and a lot of concepts starting with non-elementary terms. Unfortunately, I have not found

sufficient focus on a well-defined approach to information that starts from elementary set theory (2), as is common in mathematics. This would allow to precisely define both elementary and complex information (after extension by nesting). This is appropriate because information is a well-defined elementary and extensible concept. It is also important for reproducible digital encoding of "comparable information", "similar information", "searchable information".

My research did not start with these basic considerations. Originally, I simply needed a solution for an everyday application: At work, we routinely want or need to share accurate information. In medicine, the detailed experiences of colleagues are necessary to find the best treatment for a patient. Based on the patient's record and current diagnoses, we need to make important decisions about further diagnostic steps and about treatment. Of course, information about experience in patients with comparable and similar findings is helpful in making the best decisions. The important findings contain precise objectifiable quantitative information that must be uniformly defined for comparability and searchability.

However, we can only search for language-based information globally in a feasible way, using (well-known domains like) language vocabulary and specialized vocabulary with abbreviations and codes from globalized standards such as ICD-10 [22] or ICD-11 [23]. However, this is only a rough interpretation of the patient's detailed original findings, which contain a lot of measurement data.

It was clear to me that the global search and comparison of quantitative data should be practicable. Modern diagnostic devices automatically provide precise and meaningful measurement data from patients. All these medical data are nothing more than sequences of numbers—like all digital information (Figure 1). So I looked for digital ways to define and share sequences of numbers, such as numbers that publish quantitative data on the Internet. Initially, I looked for data from medical reports for comparison and decision support. That motivated me a lot. In orthodontics, for example, we could then define (the coordinates for) different key points in the dentition and jaw globally, so that the software of new 3D intraoral scanners could be designed to calculate them automatically. This software could then secondarily calculate other measurements automatically, such as angles and distances, as in Figure 2. We could define these data globally and share them with each other if desired, or as anonymized statistics to protect privacy. Statistics on long-term treatment results could then be generated immediately using numerical search commands—for decision support.

Of course, for global numeric search the searched number sequences should have globally uniformly the same definition and thus also the same domain.

### 4.3. The Domain of Information Is Essential

The domain is an essential aspect of information. It is a *common* ordered set from which the (digital) information (number sequence) is selected. Its definition *must be clear* for the software and the participants of the communication. If we consciously and systematically take this fact into account from now on, we can systematically increase the range and possibilities of information exchange. This is possible by means of the online definition of adapted domains, as described in Section 2. The defined data (1) are precisely comparable and searchable. Until now, the searchability of digital information has been limited to a few well-known domains such as language vocabulary. This may be obvious at first because it is adapted to our prior knowledge and brain. Anyone who speaks a particular language knows its vocabulary. Its elements (words and phrases) are combined in many ways to form language-based information. These semantic combination possibilities are so diverse that they are sufficient for encoding everyday communication but this variability also means that encoding by language is not reproducible (and comparable). Moreover, it was established long ago that language vocabulary as a domain is not sufficient to convey precise information. The elements (words and phrases) are too imprecisely defined and their resolution was insufficient. Therefore, numbers were introduced as information carriers. They have large "quantitative" domains whose order and resolution can be defined

individually and variably in the local context. There are many possibilities for this as well, so the global reproducibility of the encoding of quantitative information is missing here as well. This is already shown by the fact that the general components of digital information (number sequences) have not been globally searchable for decades.

From a technical point of view, however, the definition of number sequences and their domains was solved and the associated precise data search [20] can be demonstrated [14]. This is also relevant for fundamentals such as reproducible digitization and transport of information [12].

In computer science, however, other approaches to global data transport have been preferred so far (e.g., [15–18]), which also have a different focus. Emphasis is on a tabular or text-based data format with metadata.

### 4.4. About Existing Concepts of Information—Rethinking Is Necessary

Existing concepts have grown historically. The text-based input format has always been very useful in computer science, e.g., for programming. It has contributed significantly to the construction of the World Wide Web on the Internet as HTML format. Simple text editors are sufficient to enter and modify programs and web pages. They can also be used to enter data and metadata. The character codes used (ASCII and later Unicode [21]) are globally defined, clear and can be globally represented by software. Therefore, it was initially obvious to also use text-based formats such as XML (and more) for data transport. With these, data can be combined in many ways with abbreviated definitions in metadata, in a readable form.

But here lies an important root of the problem: even for the same application, there is no unambiguous reproducible conversion of original information into its digital (i.e., numeric) representation (as described in Section 2 (a)–(f)). There are many ways to define and combine the (formats, definitions and domains of) digital data (number sequences) locally. This leads to a non-reproducible digital representation. Therefore, data exchange between different systems often did not work reliably. In order to solve such "interoperability problems", the development of various standards was started decades ago [15]. This was not sufficient, therefore separate standards were developed for important application areas, e.g., healthcare [24,25]. The effort was and is considerable. There are now many local applications for these standards. Unfortunately, the root of the problem has remained. To this day, there is a lack of reproducibility [12] of digital information representation. The consequences of this lack of fundamentals are many. Without reproducibility, solving interoperability problems is patchy. A universal search engine for quantitative (e.g., medical) data [14] is not globally extensible due to the lack of globally defined data (1). Despite decades of the Internet, there is still no practicable way to globally define digital data (sequences of numbers with domains).

So what could be the reason for ignoring the efficient technical solution (1)? What are the disadvantages of the DV data structure?

### 4.5. Disadvantages of the New DV Data Structure

- Today still unfamiliar even for experts

The new DV data structure involves a fresh start in thinking and is developed and optimized independently. This is a strength, but it can also be a hindrance at first if the motivation for such a new start is lacking because a lot has been invested in established structures. This may explain why many decision-makers have so far preferred to ignore this—despite clear arguments.

- Initial effort

It requires some initial effort to create accurate standardized online definitions (as opposed to quick and crude local definitions). If the goal of communication is not to convey precise comparable facts (e.g., in simple and quick everyday communication), the use of online-defined DVs may seem unnecessary and initially less adapted to the human brain

than a well-known human language. However, it may also be a goal of future research to create online definitions of DVs (with simple pronunciation, see below in Section 4.6) that are more convenient and easier for the human brain to use.

- Online definition is unusual up to now and easy to underestimate

The global definition of number sequences (as components of digital information) has not been focused on in computer science education so far. Therefore, their potential is easily underestimated or not recognized at all.

- Large project which initially requires investment

The introduction of the new DV data structure is a large project that initially requires investment: The infrastructure for the new DV data format must first be established. Internet sites with online definitions must be programmed. These must provide the online definitions in a standardized, machine-readable format and mirror them for backup. Existing online definitions cannot simply be deleted but can be marked as "obsolete". Software must be developed to handle the new data format, such as DV editors and professional global DV search engines that automatically use the online definitions.

Despite the associated costs, it is plausible that the significant advantages of the DV data structure (1) listed in the next Section 4.6 will become apparent and automatically make the investment in further software attractive—as soon as the DV data stock is sufficiently large and thus attractive.

### 4.6. Advantages of the New DV Data Structure

The DV data structure is universal, fundamentally renewed and optimized for digital information exchange. This results in many advantages, which also show promising possibilities for new research.

- Global digital information including definition

The following question accompanies any digital information exchange:

How is it ensured globally that every communication participant with software knows the format and definition of the exchanged digital information (number sequence)?

This question is of central importance for digital information exchange and requires a clear answer. The DV data structure provides such an answer globally. With this, the DV requires no or only minimal metadata because the UL refers to the globally unique online definition. This definition is machine-readable, so the format of each number is immediately clear and the user's software can automatically download the relevant parts of the online definition. Depending on the state of the standard, the online definition can become more and more detailed (see Section 2 e) to enable comfortable handling of DVs by programs (e.g., search engines, editors, specialized software).

- Global digital information including identification

The UL is also an identifier for a uniformly defined group of DVs. This can be used, e.g., for global comparison and precise search within this kind of information.

- Globally reproducible and precise digital information

See Section 2: The online definition of DVs with adapted domain [9] unambiguously describes a one-to-one conversion of the application-specific (domain of) original information into (the domain of) a number sequence. This enables a reproducible digital information transport of original information. Thereby a precise information representation is possible. The accuracy of the (domain of the) numbers in the DV can be adapted to the requirements.

- Globally comparable and searchable digital information

Typically, the defined numbers that represent application-specific features are also the ones that should be comparable and searchable. The users can define all features which are of interest to them. Search is possible by one and the same universal numeric search engine (Figures 3 and 4).

For evaluation purposes, not only distance functions but also other multidimensional functions of the numbers can be defined (e.g., functions to search for a specific area or volume).

- Global digital information representation optimized for the application

As described [9], the online definition focuses on and quantifies relevant application-specific features so that the DV represents these features without redundancy. Figures 3 and 4 show a simple example.

- Global language-independent information transport

The domain of a DV is primarily adapted to the application. This adaptation is language-independent, as long as the online definition of a DV is multilingual possible and translatable.

- Digital information for global AI and machine learning

Obviously, the potential of AI depends strongly on the data to which it is applied. For targeted AI and machine learning, application-specific features are of interest. In medicine, for example, these include diagnoses and findings, treatments and outcomes. Machine learning on these data can later help for the prediction of treatment outcomes and for decision support. Since the online definition of DVs is globally valid, it can help to collect as much data as possible globally and prepare it for global machine learning without local bias.

- Digital information for global programming

The efficiency of DVs also makes them attractive for global programming interfaces. The binary self-expanding format of numbers can be made even shorter than the usual formats of program variables such as boolean, integer, float, etc.

- Digital information for efficient global work-sharing

Today we have extreme digital redundancy. For the same application, the software is often developed again and again, with different user interfaces. Thereby the data are also redundantly defined and therefore not comparable.

In contrast, the DV is designed for efficient work-sharing, because for the same application globally only one online definition is needed, which can be optimized. If there are multiple definitions for the same application, they are not hidden. They can be compared online, e.g., in terms of the first date and frequency of use.

- Digital information optimized for efficiency

Even a detailed online definition does not hinder the efficiency of information transport, because the transported and defined DV contains only the (short) UL as a pointer to the online definition. The data structure of the DV (1) is optimized for uniqueness and efficiency. It avoids detours. All numbers (components) of the DV can have a self-expanding binary structure. For example, a self-expanding positive integer can start with 4 bits (half a byte) [10]. Such efficient encoding is suitable for saving energy and optimizing performance. Nevertheless, DVs can be combined and incorporated into popular formats (such as XML, JSON, Turtle) via RFC 4648 [26]. Editors can be customized to use the online definition automatically.

- Rethinking (digital) information

So far, the domain of information (2) has not been explicitly considered. Rather, it was implicitly assumed that the domain is clear, e.g., knowledge of letters and language

vocabulary or even specialized vocabulary that often only certain experts can understand. However, here it is explicitly stated that the domains used must be clear beforehand. We can adapt domains to the application in an optimized and globally consistent way by defining them online, thus gradually extending the global domain of digital information. In doing so, we are automatically reminded of a precise, reproducible representation of the application-specific features in the domain of information, see Section 4.3.

- Semantics (combination of digital information) are better reproducible

Similar to words of a language, DVs can be combined, e.g., with "not", "and", "if", "or", "then". This combination can be shorter and therefore better reproducible since each DV usually represents more complex and precise information than a word.

When previous online definitions are reused, combined, and nested within online definitions, this is fully globally reproducible due to the global validity of the online definition.

- Improving international communication

The language-independent and globally defined DV data structure can become an important means for improving international communication to enable much better international cooperation. This is more and more necessary to solve future tasks for the survival of mankind.

Language vocabulary is not a constant but changes over time. It is possible to introduce certain definitions of DVs that are so practical and convenient for people that they can become part of the language. In doing so, the online definition may even include a suggested pronunciation (such as "datetime . . . " or "dati . . . " for short). The combination of such small units would not be fully reproducible but may be much more reproducible than the current language. The definition of such DVs adapted to the human brain could become a new topic for future research.

DVs can be used efficiently and universally as digital information due to their data structure (1). Their globally uniform definition thus results in many advantages, only some of which can be listed here.

*4.7. This Has an Impact on My World View*

As reminded in Section 2, information is always a selection from a domain (2). This domain of information must be clear beforehand. Information builds on domains that are already clear. This is an aspect that, upon closer examination, even has an impact on my worldview. You may be interested to know why.

Information is essential for our consciousness. Now, what does the "domain" of information mean in this context? The domain must be clear, i.e., we must have learned (i.e., decided or perceived) it in the past. The "language vocabulary" domain clarifies some details about "conscious" and "unconscious" information domains. These domains are "familiar" or "quickly remembered"—the faster, the more "unconscious" (the memory or) access. This is the reason why we can use language vocabulary in a practical way. We learned such relevant unconscious domains early in childhood. However, the vocabulary of language is learned later than other more upstream domains that our brain had to learn in order to deal with our body, our senses, and the environment. Early learning of important (even essential) domains of information is a prerequisite for later practicable (fast, unconscious) handling. Roughly speaking, the earlier the domain was learned (decided or perceived), the faster the access is.

Now the question arises, which information domains are even further upstream? How far upstream can we go?

I was impressed that I could find no exception to rule (2). It reaches into the domains of elementary physics. It is a prerequisite for information transport after any physical measurement [19]. Thus, any electronic (and also other everyday) communication is possible only because there is a clear common (domain of multiples of the) elementary charge. There are also other essential common domains, e.g., common multiples of elementary particles. This speaks for the fact that there must be a location-independent and even

further upstream primary domain as a common minimal precondition for the information exchange. This primary domain must be globally referenced at an extremely high frequency, of which we can only measure or "perceive" a very small part locally.

**Remark.** *To ensure consistency, access to the primary domain must be possible location-independently (globally) along any advance of time (i.e., maximally fast). This and further considerations, e.g., about overlaps of geometrical and statistical formulas and constants suggest to me that the geometrical appearance of space (of the medium for information transport) is only a (due to the limited speed of light resp. information) delayed statistical consequence of an information-theoretical combinatorial law which starts from the primary domain with extremely high frequency.*

It turns out that this domain and its associated combinatorics are an interesting topic for new research in physics. My approach here is (of course) incomplete.

However, with respect to digital applications, the approach is complete and also demonstrable [14]. Online, we can define globally uniform domains of digital information. The advantages of this application are enormous and have not yet been utilized.

### 5. Conclusions

Since information is a universal concept that is also the basis of research, the most important limit for research on its exact global definition is only given by the limits of human intelligence itself. This is also evident when thinking about upstream physical domains in the previous section.

As an application, the Internet enables the online definition of globally uniform domains of digital information. Thereby the digital information (number sequence) can be efficiently represented by the universal DV data structure (1). It is identified and defined globally uniformly. Its machine-readable online definition allows quick access to all necessary information about the format and meaning (domain) of each digital number. The online definition can be adapted to the application independently of the language (multilingual) so that the DV represents the application-specific, reproducible features in a precise, comparable and globally searchable way.

In simple and fast everyday communication, the use of online defined DV may seem unnecessary and, to date, less adapted to the human brain than a well-known human language. However, if the goal of communication is to convey precise comparable facts, as in professional communication, e.g., in science, medicine, industry, and business, the introduction and increasing use of the online defined DV data structure (1) has relevant advantages and is therefore recommended.

**Funding:** This research received no external funding.

**Institutional Review Board Statement:** Not applicable.

**Informed Consent Statement:** Not applicable.

**Data Availability Statement:** Not applicable.

**Conflicts of Interest:** The author declares no conflict of interest.

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
