# Peer review of "We Can Define the Domain of Information Online and Thus Globally Uniformly"

_information, doi:10.3390/info13050256_

Round 1
Reviewer 1 Report
In the article, the author introduces of the Domain Vector (DV) data structure: "UL plus number sequence", which uses numerical transformations to enable data to represent application-specific features in an accurate (one-to-one), comparable and globally searchable manner. This has great technical potential for the future of computer science. The problems presented in this paper and the solutions to them are novel, and the evaluation of the proposed methods is comprehensive.
I have the following additional suggestions for the paper.
(1) The article describes the application background of orthodontics at the beginning and the end, but in the main body of the article, there is no connection between the proposed method and the application background. It does not allow the reader to know how the method proposed in this paper applies to orthodontics.
(2) The article mentions "New DV data structure", but only describes it briefly , without specifying the theory and analyzing the experimental results, which can't let the readers understand how to achieve "global definition of information domain".
(3) The conversion of raw information into digital sequences is mentioned several times in the article, how does the conversion process work? I don't see it described in the article.
(4) The article only shows some of the author's ideas, and does not clearly demonstrate the author's innovative points.
(5) The meaning of some sentences is not clearly understood by the reader and needs further revision in terms of writing.
Author Response
Thanks for the comments.
My answers are contained in the uploaded word file.

Reviewer 2 Report
The paper has to be improved accordingly:
- The title of the paper is too wide. It should be reformulated as the author does not touch the Morse alphabet, information coding with hands gests.
- The introduction should not include figure, it is better to provide here definition of the domain of information and links with hybrid domain, cross-domain.
- By the end of the introduction the structure of the paper could be introduced.
- The author has to include contemprorary publications from 2021-2022 years. It is suggested to extend literature review.
- The methodology is not presented in the paper. The author could show research structure and announce research questions.
- The figure 1 could be presented in the research object description part, it could be integrated into the methodology section.
- The author has to provide discussion section and show here what is researched by other authors and what are the gaps that are filled with this paper.
- By the end of the discussion section the author has formulate further research directions.
- The author has to extend the conclussions section and provide research limitations here.
- The abbrevation has to UL has to be explained in the text when it is first mentioned.
Author Response

(The authors gave the same response as above.)

Reviewer 3 Report
In this paper, the author discussed the Domain Vector (DV) data structure: “UL plus number sequence” for digital information processing. This manuscript is more like a communication letter other than a research article. The author almost reviewed his own papers and his thought about digital information processing more ten years ago. Then the author recommended the definition of digital information. However, no scientific method was developed, and no experiments demonstrated the DV data structure is useful. It is not clear what are the methodically contributions.
Author Response

(The authors gave the same response as above.)

Round 2
Reviewer 2 Report
Thank you for the paper improvement. I have no comments for the author.
Reviewer 3 Report
The author almost rewrote the paper. The paper now is well structured in more logic way. In particular, the author added the basic method section and presents some preliminary results from the search results to demonstrate his research idea. The Discussion section is deep and rich. The quality of the paper has been improved a lot.